# Preliminary Results of Feasibility and Acceptability of Self-Collection for Cervical Screening in Italian Women

**DOI:** 10.3390/pathogens12091169

**Published:** 2023-09-17

**Authors:** Illari Sechi, Narcisa Muresu, Mariangela V. Puci, Laura Saderi, Arcadia Del Rio, Andrea Cossu, Maria R. Muroni, Santina Castriciano, Marianna Martinelli, Clementina E. Cocuzza, Giovanni Sotgiu, Andrea Piana

**Affiliations:** 1Department of Medicine, Surgery and Pharmacy, University of Sassari, Padre Manzella 4 Street, 07100 Sassari, Italy; illasechi@uniss.it (I.S.); andreacossu@uniss.it (A.C.); mrmuroni@uniss.it (M.R.M.); piana@uniss.it (A.P.); 2Department of Humanities and Social Sciences, University of Sassari, Padre Manzella 4 Street, 07100 Sassari, Italy; 3Clinical Epidemiology and Medical Statistics Unit, Department of Medical, Surgical and Pharmacological Sciences, University of Sassari-Padre Manzella 4 Street, 07100 Sassari, Italy; mvpuci@uniss.it (M.V.P.); lsaderi@uniss.it (L.S.); gsotgiu@uniss.it (G.S.); 4Biomedical Science PhD School, Biomedical Science Department, University of Sassari, Padre Manzella 4 Street, 07100 Sassari, Italy; delrio.arcadia2@gmail.com; 5Copan Italia SpA, 25125 Brescia, Italy; santina.castriciano@copangroup.com; 6Department of Medicine and Surgery, University of Milano-Bicocca, Cadore 48 street, 20900 Monza, Italy; marianna.martinelli@unimib.it (M.M.); clementina.cocuzza@unimib.it (C.E.C.)

**Keywords:** cervical screening, vaginal self-collection, human papillomavirus, acceptability self-collection, HPV-DNA test, elution medium

## Abstract

Background: Given the diagnostic accuracy of HPV-DNA tests in terms of self-collected samples, in order to implement self-sampling in cervical screening programs, the standardization of the pre-analytical phase, including decisions concerning the choice of medium, the volume of elution, and storage conditions, are necessary, in addition to understanding the potential factors involved in acceptability by women. On this basis, we carried out a cross-sectional study to assess (i) the stability of dry vaginal self-collected samples stored at room temperature for up to 4 weeks after elution in 2 mL of eNat^®^ (Copan) medium, and (ii) the acceptability of self-collection in enrolled women. Methods: 185 women were enrolled in the LILT (Italian League Against Tumors) regional project. A self-sampling kit, including a dry FLOQSwab^®^ (Copan), instructions for use, and a satisfaction questionnaire, were supplied for each woman and sent by mail to the laboratory. The HPV-DNA test was carried out using the Anyplex™ II HPV HR (Seegene) kit. To evaluate the specimen’s stability, 185 dry vaginal swabs were eluted in eNat^®^, a lyses-based molecular medium and tested for HPV detection at two different time points (<6 days and 1 month after elution). The Cohen’s Kappa coefficients and McNemar test were used to assess the agreement of HPV-DNA at different times. Results: We found high agreement in terms of HPV-DNA results among the samples tested at two different time points (Cohen K = 0.98; *p* < 0.0001). Moreover, most of the women found it easy to use self-collection devices and the pictorial instructions clear to understand. Approximately half of the enrolled women declared preferring self-sampling to clinician-collected methods. Conclusion: Our results display the high reliability and accuracy of HPV-DNA tests using dry vaginal self-collection FLOQSwabs^®^ devices eluted in 2 mL of molecular medium. The analysis of the questionnaire showed a high acceptability of self-collection among women, although a high percentage preferred standard collection devices. Overall, our preliminary results support the adoption of self-collection in screening programs, even though further analyses should be performed to optimize and standardize protocols for HPV tests on self-samples, and educational campaigns are needed to adequately inform and increase responsiveness in a target population.

## 1. Introduction

Cervical cancer is the fourth most common cause of cancer among women worldwide, with more than 604,000 new diagnoses and over 341,000 deaths in 2020 and the third most common cause of cancer in women aged 15 to 44 years [1].

Human Papillomavirus (HPV) infection is known as the main cause of cervical cancer, with HPV-16 and -18 accounting for approximately 70% of cervical cancer. However, only a small fraction of HPV infections are persistent, whereas the majority resolve spontaneously [2]. Virtually 100% of HPV-related lesions could be prevented via primary and secondary preventive strategies (i.e., vaccination and cervical cancer screening). Through the global implementation of cytology-based screening programs, it has been observed that the incidence trend is slowing down or has even stopped, likely due to growing exposure to oncogenic genotypes, lower screening coverage and the limitations of cytology [3].

The global strategy for cervical cancer elimination by the World Health Organization (WHO) recommends primary and secondary preventive interventions, including vaccination and screening: vaccination for 90% of girls globally, 70% of screened women aged 25–64 years, and treatment of 90% of cervical disease by 2030 [4]. On these bases, international guidelines recommend the adoption of HPV-based screening, which provides several advantages, such as greater protection than cytology against invasive cancer, prolonged intervals for follow-ups and starting at 30 years of age [3].

Currently, cervical screening shows relevant geographical variations in terms of organization and coverage. Despite the available screening modalities, a recent review revealed that two in three women aged 30–49 years have never been screened for cervical cancer globally. Moreover, a different distribution was reported by income level, with coverage being nine times higher in high-income countries than in low-income countries (9% vs. 84%, respectively) [5]. The discrepancies in screening coverage reflect the incidence and mortality ratios for cervical cancer, with ~80% of cases reported in low–middle income countries. In addition, cytology-based programs are particularly difficult to implement in low-resource regions since they require trained personnel to obtain adequate smears and to perform a correct interpretation of results, suggesting the adoption of alternative preventive strategies [1]. To this, HPV-DNA-based testing is recommended as the first choice in primary screening programs due to its sensitivity, cost-effectiveness, and potential application in self-collection.

The screening strategies used in Italy slightly vary among regional organizations: on average, ~77% of women aged 25–64 years participate in cervical screening through organized programs or spontaneous initiatives. Socio-economical and geographic differences were registered at the national level: the participation is higher among socio-economically advantaged women, associated with economic status and educational level, higher in those who are married or cohabiting and lower in foreign women. Moreover, a north–south gradient is observed in the screening uptake, with a percentage ranging from 91% in North Italy to ~65% in the south. A significant fraction of women (~25%) report that they have never undergone cervical screening or are screened regularly, and the main reason is the low individual risk perception [6]. In Sardinia, an organized screening program is available, with an active call for the targeted population, based on pap smears as the first level, followed by HPV-DNA PCR testing for women with cervical dysplasia at cytology examination. Recently, due to the COVID-19 pandemic, interventions for cervical cancer prevention have slowed down, either in terms of primary preventive strategies (i.e., vaccination) or screening adherence, with a decline in invitations and propensity to participation.

In this scenario, it has been emerged the need to implement knowledge about HPV-related diseases and prevention systems, with health education intervention, and, contextually, considering new screening strategies, particularly addressed to non-adherence women and women with prior treatment for cervical high-grade lesions to reduce the risk to develop HPV-related extra-cervical cancers [7,8].

The good analytical performance of HPV-PCR testing in self-collected samples, along with the high acceptability by women due to its ease of use, safety, privacy, and convenience, supports its adoption in regular screening programs as a complementary choice [9,10,11,12]. However, numerous studies are currently evaluating what might be the most effective approaches for large-scale implementation of self-collection in different geographic and socioeconomic contexts [13]. In particular, the higher sensitivity of HPV-DNA PCR testing in relation to vaginal self-collection should be affected by the pre-analytical steps. Several studies highlighted the need to review the current protocol for vaginal self-samples, particularly those regarding the validation of specific cut-offs in HPV testing and specimen adequacy, in order to reduce the number of inadequate or invalid tests. Among factors that could affect the accuracy of HPV testing, the choice of transport medium, the adoption of wet or dry swabs and the volume of the sample elution medium, ranging from 20 mL to 1 mL, have been reported [14]. Standardized procedures are needed to optimize HPV tests on self-samples, ensuring diagnostic accuracy on a par with clinician-collected samples. Moreover, it has been demonstrated that the acceptability of self-sampling highly depends on individual beliefs and knowledge and requires appropriate intervention to maximize its benefit against cervical cancer [15].

To answer the still open questions related to the pre-analytical phase and laboratory processing of specimens, a preliminary study based on cervical screening by vaginal self-collection was conducted in the Sardinian region (Italy). Volunteer women were enrolled within the regional LILT (Italian League Against Tumors) project, a non-profit organization aimed at health promotion and prevention at national and regional levels, in collaboration with the Department of Medicine, Surgery and Pharmacy of University of Sassari (Sardinia, Italy), member of the Global HPV Laboratory Network by WHO for HPV-DNA testing. Self-collection kits were provided to the recruited women, aimed to assess (i) the pre-analytical performance of dry vaginal self-samples eluted in 2 mL of molecular medium and analyzed using a HPV-DNA PCR test at two different time points, (ii) the acceptability of self-collection devices in relation to the enrolled women.

The results obtained will provide crucial information for the implementation of screening based on self-sampling, addressing the gaps regarding the pre-analytical phase and the preparedness of the target women.

## 2. Materials and Methods

### 2.1. Study Population

A cross-sectional study was conducted between March 2021 and June 2022 in Sardinia (Italy).

The inclusion criteria for enrolled women were: (i) age from 25 to 64 years; (ii) women who did not participate in cervical screening during the last year; (iii) women with no prior treatment for cervical cancer; and (iv) no pregnant women. Staff involved in each regional LILT center promoted the initiative through the available communications channels, such as social communities, media, newsletters, or individual contacts. Volunteer women who met the selection criteria were recruited for the study and invited to pick up the self-sampling kit and to sign the written informed consent at the LILT territorial office.

All of the performed procedures were in accordance with the ethical standards of the local Ethical Committee and with the Helsinki declaration. All women were informed by qualified health workers about the study objectives.

### 2.2. Samples Collection and Processing

The self-sampling kit provided to the enrolled women contained a dry vaginal swab with illustrative instructions for self-collection, directly provided by the self-collection device manufacture’s and a questionnaire to collect demographic data and specified questions to assess the perceptions within the study population about vaginal self-sampling and the device’s ease of use. Vaginal self-sampling was collected using dry FLOQSwabs^®^ 552C.80 (Copan Italia, SpA, Brescia, Italy), a flocked swab that consists of a molded plastic applicator stick with a regular and shaped tip. The tip of the applicator is coated with short Nylon^®^ fibers that are arranged in a perpendicular fashion. The samples collected and questionnaires were returned to each territorial LILT center and delivered to the reference laboratory through the healthcare practitioner.

At the Molecular Epidemiology Laboratory, all the dry vaginal FLOQSwabs^®^ were first soaked in 2 mL of eNat^®^ (Copan Italia, SpA, Brescia, Italy) medium, a lysis-based molecular medium, and then vortexed for at least 30 s. Subsequently, the swab was removed and discarded. Lastly, the eluted samples were analyzed for HPV-PCR at two different time points in order to verify the stability of media stored at room temperature for up to 4 weeks.

The eNat^®^ medium is a commercially available guanidine thiocyanate-based molecular medium that stabilizes nucleic acids (i.e., DNA and RNA) at room temperature for over a month, which degrades proteins and inactivates microbial infectivity through the lysis of the cell membranes. The inactivation is due to the synergistic interaction of guanidine thiocyanate and detergents. Dry vaginal self-samples were resuspended in 2 mL of eNat medium and processed as follows.

A 200 µL aliquot of eNat^®^ (Copan Italia, SpA, Brescia, Italy) medium tube was used for DNA extraction by the Seegene AdvanSure™ E3 SYSTEM [16]. This automatized system allows for the extraction of DNA from human specimens by using magnetic beads coated with iron oxide. Lysis buffer, washing buffer and elution buffer are loaded into each well of the reagent plate. The high concentration of Chatropic salt contained in Lysis buffer destroys the cell membrane and separates DNA from protein. In this environment, the magnetic beads absorb a nucleic acid with a negative charge. The DNA absorbed by the magnetic beads is moved sequentially toward each column of the plate and passes through several washing steps using the magnetic rod of a nucleic acid extractor. In the end, elution buffer separates the DNA from an absorbed status of the magnetic beads, and once the magnetic rod removes the magnetic beads, only the DNA acid remains in the elution buffer. The eluted DNA was immediately analyzed for the detection of HPV-DNA.

In order to assess the stability of the medium over time, a second 200 µL aliquot of eNat^®^ medium was stored at room temperature for ~30 days and, subsequently, subjected to a second DNA extraction and HPV testing.

All HPV-DNA PCR tests were carried out using Anyplex™ II HPV HR (Seegene^®^ Inc., Seoul, Republic of Korea) kit using real-time PCR [17]. This technology allows the detection of 14 high-risk HPV genotypes (HPV-16, -18, -31, -33, -35, -39, -45, -51, -52, -56, -58, -59, -66, and -68) using a single fluorescence channel on real-time PCR instruments for multiple genotypes. The amplification of HPV DNA was performed on the CFX96 thermocycler (Bio-Rad), according to the manufacturer’s instruction. Simultaneously, a fragment of a house-keeping gene was amplified during the reaction to ensure the presence and adequacy of cells and the absence of PCR inhibitors. The Anyplex assay offers a semi-quantitative result using a crossing point (Cq) range: 31 or fewer cycles (+++), from 31 to 39 (++) and more than 40 cycles (+). Samples were considered invalid if the internal control had a Cq greater than 40 and was negative for HPV.

### 2.3. Questionnaire for Epidemiological Data Collection and Acceptance of Self-Sampling

The questionnaire (Appendix A) collected demographic information about age, civil status, educational level, smoking status, BMI (Body Mass Index), menopause status, number of pregnancies, use of contraceptives and/or condoms, and previous sexually transmitted infections (STIs). Some questions have collected clinical data regarding the frequency of participation in cervical screening and the results of previous tests. Finally, three questions were focused on vaginal self-sampling acceptability and ease-of-use for participants, with answers given as “yes” or “no”. The following questions were included:▪Did you find the self-collection instruction clear and understandable?▪Did you find self-collection easy to perform?▪Do you prefer using the vaginal self-collection than the clinician-collection device?

### 2.4. Statistical Analyses

Demographic (i.e., age, level of education, civil status, menopause status, exposure to contraceptives, use of condoms) and clinical (i.e., data and result of the last cervical screening) variables were summarized with median and interquartile range (IQR) for quantitative variables, and with absolute and relative (percentage) frequencies for qualitative variables. Shapiro–Wilk test was used to assess the normality of the data distribution. The McNemar test was used to evaluate the differences between HPV-DNA results at different time points. Cohen’s Kappa coefficients were calculated to assess HPV positivity agreement. The statistical significance threshold was set at <0.05, and data analysis was carried out through STATA version 17.

## 3. Results

A total of 185 females were enrolled during the study period, with a median (IQR) age of 41 (34–51) years. Most women (39.1%) had a high level of education, nearly 60% were married, and 24% were in menopause. Over ninety-five percent reported use of contraceptives during their lifetime, whereas 23% never used a condom (Table 1). Twenty-eight (15.1%) never attended any cervical screening, and over 62% underwent their last pap-test more than three years before.

In this study, 37.2% were smokers, and ~8% of women reported a history of previous sexually transmitted infections (STIs).

Overall, 17.3% (32/185) reported positivity for at least one high-risk HPV genotype. The most common HPV genotype detected was HPV-56 (16.3%), followed by HPV-18, -31, -51 (12.2%) and HPV-52 (10.2%) (Figure 1). The majority of HPV-positive cases (62.5%) were associated with a single HR-HPV genotype. None of the samples were invalid following the HPV-DNA assay.

### 3.1. Stability of Vaginal Self-Collected Samples Eluted in eNat^®^ Medium

Dry vaginal FLOQSwab^®^ (n=185) samples were stored at room temperature for a median (IQR) of 19 (13–32) days before resuspension in 2 mL of eNat^®^ medium. The first HPV-DNA test (t0) was carried out after a median (IQR) of 3 (2–6) days after elution, with a positivity of 17.3% (32/185). The second HPV-DNA test (t1) was carried out after a median (IQR) of 35 (35–39) days after the first analysis (t0). The results at t1 confirmed the positivity at t0 for most samples (16.8%, 31/185), whereas all of the samples that tested negative at first analysis were confirmed at t1 point. The results for HPV-positive specimens at t0 and t1 are reported in Table 2.

Overall, the rate of “invalid” results was 0%, and no statistically significant differences in HPV positivity were found between t0 and t1 (17.3% vs. 16.8%; *p*-value: 0.32), with an agreement of 99.5% (*p*-value < 0.0001) (Figure 2).

### 3.2. Results of Acceptability Questionnaire

Regarding the acceptability questionnaire, the majority of women found the instruction of self-collection clear and understandable and the collection easy to perform, while 97% of women answered positively to the question “Did you find the self-collection instruction clear and understandable?”, whereas 95% of “yes” answers were registered for the second question “Did you find self-collection easy to perform?”. Conversely, 45% of participants declared to not prefer vaginal self-sampling to clinician collection (Figure 3).

No significant differences were recorded in the questionnaires based on the different demographic (i.e., civil status) and epidemiological (i.e., number of pregnancies, use of contraceptives) variables. However, although below the limits of statistical significance, a higher level of self-sampling acceptance (3rd question) was observed among women younger than 41 years old and those with an advanced level of education (percentage of acceptability higher than 60%). Moreover, a slight geographic difference was found in the level of acceptability among the female respondents, with a higher preference for self-collection among women living distant from the principal regional screening centers (69% vs. 45% of “yes” answers to the third question). No difference in the questionnaire results was found between “non-adherence”, or not regularly receiving screening, to screening programs and those who regularly underwent pap smears.

## 4. Discussion

Vaginal self-collection is a safe and reliable method to involve women in cervical screening programs, showing similar diagnostic accuracy to clinician-collected samples [5,18]. Despite the fact that the WHO recommends primary HPV-based screening and approves the introduction of self-collection in screening programs, several shortcomings related to the pre-analytical phase of processing, such as transport, storage, and laboratory handling, need further investigation. Moreover, potential failures in knowledge of and reliance on the new collection device should represent barriers in the implementation of vaginal self-sampling-based screening. The preliminary results observed in our study highlighted the reliability of HPV-DNA testing carried out on self-collected specimens using a dry vaginal FLOQSwabs^®^ (up to ~30 days) and eluted in 2 mL of eNat^®^ molecular medium, suggesting a pre-analytical protocol for the workflow of self-sampling.

Although several collection devices have been validated in recent years for use in vaginal self-collection, the designated population, climatic and geographical conditions, and logistical infrastructures are essential elements to consider prior to adoption [19]. Dry vaginal swabs are a considerable solution to prevent the risk of spillage during transport but, increased attention is required concerning the choice of resuspension medium and volume.

The stability of dry vaginal self-samples in eNat^®^ medium, as highlighted by the high agreement in terms HPV testing at different time points, addresses one of the current needs for the implementation of self-collection in cervical cancer screening programs, particularly those related to transport and stability of specimens. Moreover, the reduced volume of eNat^®^ medium (2 mL) makes it possible to increase the concentration of the sample, avoiding the risk of ‘invalid’ or false negative results and allowing safe transport. In fact, recent studies demonstrate that the resuspension of dry vaginal swab in 5 mL or less may maximize the detection of the HPV genome [19].

In addition to elution volume, another relevant feature is the composition of the media. Until now, cytology alcohol liquid-based transport medium, usually employed for the transport and storage of cervical samples, has several disadvantages (i.e., costly, flammable, toxic and not suitable for mailing) [20]. The use of a non-alcohol-based medium can overcome these issues, enabling samples to be shipped remotely without affecting the stability of the DNA and RNA.

Besides preserving and stabilizing the nuclei acid for molecular analysis for up to one month, the eNat^®^ medium allows the inactivation of microorganisms in clinical samples and, contextually, reduces the risk of exposure to infectious agents for laboratory workers [21,22].

The questionnaire results showed that the majority of participants considered the self-collection procedures easy to perform and that the instructions provided were clear and understandable. Our results confirmed previous data about women’s acceptability of self-collection. Women mainly appreciate the “easiness”, “rapidity” and “comfort” of the device, as well as the possibility of overcoming the embarrassment and emotions involved with a visit to the gynecologist [23]. All these reasons contribute to the choice of this prevention tool for “hard-to-reach” women who do not or not regularly participate in organized screening programs. Overall, we observed that a significant percentage of women declared preferring clinician- rather than self-collection methods, confirming that a relevant percentage of women are still hesitant about the use of new collection devices. Similarly, previous population-based studies have demonstrated lower adhesion to self-collection when standard clinician sampling was available [24]. Moreover, a recent meta-analysis found that lack of knowledge or awareness, negative perception of testing (e.g., fear and anxiety), and low accessibility to services are the main reasons behind poor adherence [25]. Accordingly, we observed higher acceptability in women with higher levels of education and among those who live distant from the main regional screening centers and, as a consequence, have more difficulty in accessing health services. This phenomenon could be overcome through appropriate communication and educational interventions to increase knowledge and confidence about the accuracy of HPV testing on self-sampling and to maximize the benefits of novel preventive strategies [15].

These preliminary results strongly support the implementation of primary and secondary preventive procedures against cervical cancer. Recognizing the diagnostic accuracy of molecular testing for HPV detection carried out on vaginal self-collection compared with clinician-collection samples, as previously demonstrated [11]; the results obtained are roughly consistent with previous epidemiological data. In fact, the most common genotypes detected were formerly identified as prevalent in our region and at the national level, particularly for HPV-18, -31, -56, -51, and -52, supporting the adoption of the nonavalent vaccine as the primary preventive strategy [26,27]. The overall positivity found in our cohort probably reflects the wide variability in the prevalence of HPV infection reported in the literature, which is related to demographic (i.e., age), clinical (i.e., cytology and histology examinations) and microbiological (i.e., HPV detection method, type of specimens) variables [28]. The lack of information relative to cytology analysis does not allow for a comparison of the prevalence of HPV infection in our cohort with those reported previously in women with ASCUS cytology (~48% of HPV-positive rate) [27]. Further studies and larger sample sizes may clarify our results and plan appropriate preventive strategies to reach non-adherent women, who accounted for a relevant percentage in our cohort study (>60%), and to supply the consistent delay and reduction in screening participation (from 76% in 2019 to 40% in 2020), mainly attributable to the recent COVID-19 pandemic.

Although this represents the first study on self-report-based screening in our territory, it showed several limitations, mainly related to the small sample size and the monocentric nature of the study, as well as the recognized weaknesses of self-reported questionnaires. Moreover, as this project was carried out within the regional LILT program, consequently, the modality of enrollment did not allow us to apply the results to the general population, particularly those related to the acceptability questionnaire.

Overall, the adoption of dry vaginal self-collection devices used in screening could be worthwhile scaling up, not only in rural or remote regions, which register the lowest adherence to preventive programs and the greatest limitations in healthcare services, but also as a support to organized screening program. The stability of 2 mL eNat^®^ medium for the storage of vaginal specimens until one month strongly supports its potential use in cervical screening programs, especially in geographically disadvantaged areas. Further real-life research and large sample sizes are required for the standardization of screening activities based on self-collection and to improve women’s confidence.

## 5. Conclusions

The high accuracy of HPV-DNA testing in self-collected samples has been evaluated in numerous studies, although several issues, mainly related to the processing, storage, and handling of the specimens, slow down its adoption in organized screenings. The development of standard pre-analytical protocols is needed to implement this modality into a national screening program. The present study demonstrated the reliability and stability over time of HPV-DNA testing carried out with dry vaginal FLOQSwabs^®^ self-collected eluted in 2 mL of eNat^®^ molecular medium. Moreover, our results highlight that the acceptability of self-sampling decidedly depends on individual beliefs and knowledge and requires adequate health literacy intervention both for women and healthcare providers to ensure informed decision making and provide adequate support during the management of follow-ups.

## Figures and Tables

**Figure 1 pathogens-12-01169-f001:**
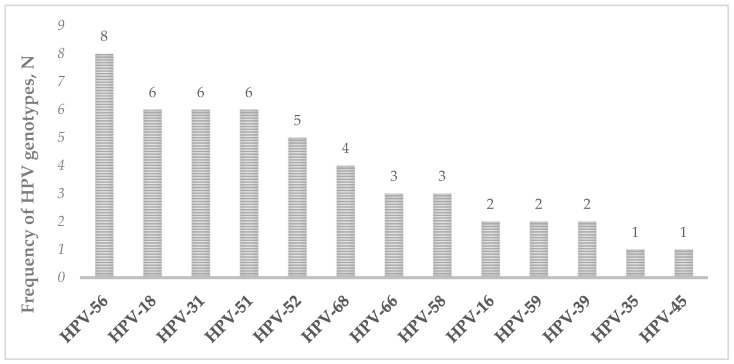
Number of frequencies of HPV genotypes detected via HPV-DNA testing in the study group.

**Figure 2 pathogens-12-01169-f002:**
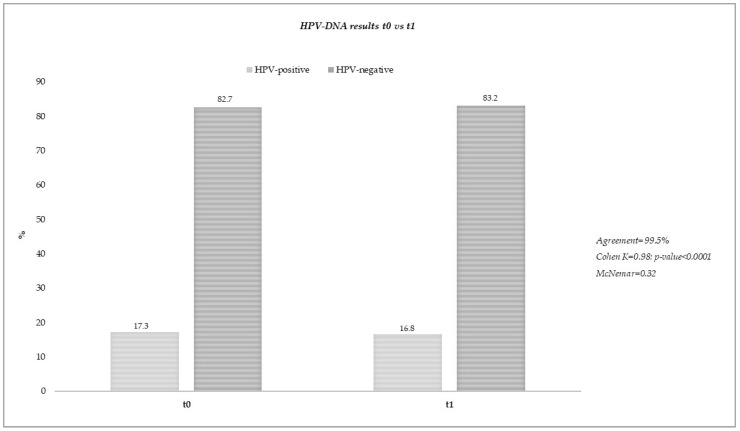
HPV-DNA test results between two different time points.

**Figure 3 pathogens-12-01169-f003:**
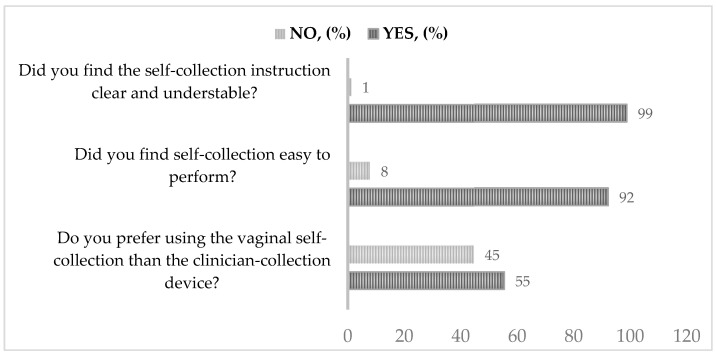
Results of questionnaire about self-sampling acceptability and ease of use among participants.

**Table 1 pathogens-12-01169-t001:** Demographic and clinical characteristics of the study population.

Characteristics		Cohort Study (n = 185)
Median (IQR) age, years		41 (34–51)
Education level, n (%)	Elementary school	0 (0.0)
	Middle school	43 (23.4)
	High school	69 (37.5)
	Degree	72 (39.1)
Civil Status, n (%)	Single	51 (29.7)
	Widow	2 (1.2)
	Married	103 (59.9)
	Divorced	16 (9.3)
Smoking status, n (%)		51 (37.2)
BMI Median (IQR)		23.0 (20.8–26.2)
Menopause, n (%)		44 (24.4)
N° pregnancy, Median (IQR)		1 (0–2)
Use of contraceptives, n (%)		146 (96.1)
Use of condom, n (%)	Never	38 (22.8)
	Always	82 (49.1)
	Occasionally	22 (13.2)
	In the past	25 (15.0)
History of STI, n (%)		14 (7.9)

**Table 2 pathogens-12-01169-t002:** Comparison of HPV-positive samples at t0 and t1.

Sample ID	HPV-DNA t0 *	HPV-DNA t1 *
SS014	HPV-51(+++)	HPV-51(+++)
SS016	HPV-56(++)	HPV-56(+)
NU026	HPV-18(++); HPV-39(+); HPV-56(++)	HPV-18(++); HPV-39(+); HPV-56(++)
NU046	HPV-31(+)	HPV-31(+)
NU048	HPV-51(+); HPV-58(++)	HPV-51(++); HPV-58(+)
NU050	HPV-52(+); HPV-56(+)	HPV-52(+); HPV-56(+)
NU051	HPV-18(+)	Negative
NU056	HPV-58(+++)	HPV-58(+++)
NU069	HPV-51(+)	HPV-51(+)
NU070	HPV-56(+); HPV-66(++); HPV-68(++)	HPV-56(++); HPV-66(++); HPV-68(+)
NU071	HPV-31(++); HPV-51(++)	HPV-31(++); HPV-51(++)
SS021	HPV-39 (+)	HPV-39 (++)
CA067	HPV-31(+)	HPV-31(+)
CA069	HPV-52(++)	HPV-52(++)
CA070	HPV-68(+)	HPV-68(+)
CA071	HPV-52(++)	HPV-52(++)
CA073	HPV-56(++)	HPV-56(++)
CA076	HPV-56(+)	HPV-56(+)
CA078	HPV-52(+)	HPV-52(+)
CA079	HPV-56(++); HPV-68(++)	HPV-56(++); HPV-68(++)
CA082	HPV-16(++)	HPV-16(++)
CA089	HPV-51(+); HPV-56(++)	HPV-51(+); HPV-56(++)
CA095	HPV-18(++); HPV-58(+)	HPV-18(++); HPV-58(+)
NU076	HPV-18(+); HPV-31(++)	HPV-18(++); HPV-31(++)
NU090	HPV-31(++); HPV-35(+); HPV-45(++)	HPV-31(++); HPV-35(+); HPV-45(++)
NU094	HPV-18(++)	HPV-18(++)
NU095	HPV-66(+++)	HPV-66(+)
NU096	HPV-51(+)	HPV-51(++)
SS053	HPV-52(++); HPV-59(+)	HPV-52(++); HPV-59(+)
SS057	HPV-31(+); HPV-66(+)	HPV-31(+); HPV-66(+)
SS090	HPV-18(++)	HPV-18(+)
SS095	HPV-16(++); HPV-68(+++)	HPV-16(++); HPV-68(+++)

* (+++) Cq ≤ 31 cycles; (++) Cq 31–39 cycles; (+) Cq > 40 cycles.

## Data Availability

The datasets used and/or analyzed during the current study are available from the corresponding author upon reasonable request.

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
