# Peer review of "Preliminary Results of Feasibility and Acceptability of Self-Collection for Cervical Screening in Italian Women"

_pathogens, 2023, doi:10.3390/pathogens12091169_

Round 1

Reviewer 1 Report

Sechi and colleagues have undertaken a self-collection study examining both the ethical and social aspects. The overall paper is good but there are a number of issues to be resolved.

- Overall there needs to be a check of grammar and spelling. There are a number of issues buta few examples "Cervical cancer is the fourth cause of cancer" (I imagine it is meant to say fourth most common or prevalent?. "Nuclei acid" and so  on

- There needs to be more detail on how patients were recruited in the methods as there is some additional information elsewhere in the manuscript but it needs to be collated. Including where the instructions for self-collection were from (are these the instructions supplied by Copan?)

There are a number of instances where the data and the explanation don't match. Including "coverage 7 times higher in high-income countries than low (9% v 84%, respectively). This is >9-fold difference. In the first paragraph of the results the numbers in Table 1, e.g. median(IQR) age is 42 (33-53) in text but 41 (34-51) in Table 1. 15% of women reported a history of previous STI in the text but its 7.9% in Table 1, and so on. It is also unclear in Table 1 what the four rows associated with Use of condom are. It would also be beneficial to know how many (if any) of the participants were vaccinated, even if the statement is that none were.

- Probably need to have a legend for the HPV genotype distribution figure including that there was no HPV33 detected, and how many infections there were.

- The manuscript refers to storing the specimens at up to four weeks at room temperature but Results 3.1 make it clear that all samples were tested for the second time 35 - 39 days (it is unclear whether this is from collection, elution, or the first test) so there is a minimum of 5 weeks and depending on what the 35 days is from could be as high as 10 weeks from collection.

- In results 3.2 there is a mention of a 'yonger (sic) cohort - what age is the cut off for this group ?

- It is probably worth separating the routine, overdue, and never screened groups for the third question in the Acceptibility questionnaire as I would imagine the never-screened would have a higher preference for the vaginal sample which could be masking the actual preference for routine screeners.

- Is there a reason (page 8) why these are only "preliminary" results?

- Probably be worth referring to HPV-DNA PCR, or even just HPV PCR, as HPV-DNA can refer to tests like HC2 and careHPV which are less sensitive for self-collection

- Please identify which Copan FLOQSwab was used, 552C.80? Also how the swab was eluted, was it snapped into eNAT, or swirled and removed (e.g. the Roche method), or something else?

- On page 4 there is a lot of information about the Seegene method. This could be reduced and other papers using this method cited instead of a full description.

- it would be good to get some context on the number of HPV positive specimens as it seems high for a screening population, and maybe even the HPV types involved compared with similar populations in this region. This may assist in context of these results in the absence of a direct comparison with cervical collected specimens.

- would suggest rewording "dry vaginal self-collection" as it may be read as "dry vaginal.......self-collection" rather than ad "dry......vaginal self-collection". Perhaps just dry self-collection as defined as vaginal earlier in the paper. 

- I could not access the supplementary materials so have no feedback on them

See above

Reviewer 2 Report

Dear authors thank you for your work. Some points should be addressed before considering it for publication:

line 75: HPV DNA based..

line 88: what do you mean? HPV testing is not the primary screening test? or is it a triage?

line 90 have slowed down

line 93: please expand about knowledge of other HPV related cancers: 1. Preti M, Rosso S, Micheletti L, et al. Risk of HPV-related extra-cervical cancers in women treated for cervical intraepithelial neoplasia. BMC Cancer. 2020;20(1). doi:10.1186/s12885-020-07452-6

line 95: also vulvar examination at the time of hpv screening or vulvar self examination might be n important moment of prevention: 1. Preti M, Selk A, Stockdale C, et al. Knowledge of Vulvar Anatomy and Self-examination in a Sample of Italian Women. J Low Genit Tract Dis. 2021;25(2):166-171. doi:10.1097/LGT.0000000000000585

line 131: inclusion criteria might bias the overall results: acceptability of self collection kit should be done on women not attending at all cervical screening. furthermore, how were the kits delivered? mailed? 

Questionnaire: is it provided as supplementary material? please provide it

not sure self reporting STIs is a reliable sort of information

line 256: true, but in your study there were not hard to reach women

Moderate editing to many typo errors and verbal tenses

Round 2

Reviewer 2 Report

I am fine with authors' review